# The Matrix Protein Cysrichin, a Galaxin-like Protein from *Hyriopsis cumingii*, Induces Vaterite Formation In Vitro

**DOI:** 10.3390/biology12030447

**Published:** 2023-03-15

**Authors:** Zhonghui Xia, Xin Zhang, Yujuan Zhou, Liping Yao, Zhen Zhang, Rongqing Zhang, Xiaojun Liu

**Affiliations:** 1China National Biotec Company Limited, Beijing 100191, China; 2Department of Biotechnology and Biomedicine, Yangtze Delta Region Institute of Tsinghua University, Jiaxing 314000, China; 3Taizhou Innovation Center, Yangtze Delta Region Institute of Tsinghua University, Taizhou 318000, China; 4Zhejiang Provincial Key Laboratory of Applied Enzymology, Yangtze Delta Region Institute of Tsinghua University, 705 Yatai Road, Jiaxing 314000, China

**Keywords:** vaterite, matrix protein, shell formation, *Hyriopsis cumingii*

## Abstract

**Simple Summary:**

Vaterite has been widely used in the pharmaceutical industry because of its large specific surface area, high solubility, high biocompatibility, low specific gravity, and other beneficial characteristics. It is an internal factor that affects the quality of freshwater pearls. The mollusk shell comprises more than 95% calcium carbonate and less than 5% organic matrix. The organic matrix of the shell includes shell matrix proteins, polysaccharides, and a small number of lipids, among which shell matrix proteins play a crucial role in the formation of shells and pearls, according to previous studies. In our present study, we cloned a novel matrix protein, cysrichin, and our in vitro experiments confirmed that the cysrichin peptide induced vaterite crystals. Cysrichin has repeat amino acid sequences and a modular structure that may play a key role in establishing a structural framework and inducing crystal nucleation and growth. We analyzed cysrichin at both molecular and protein levels and confirmed that it was an important matrix protein; this finding provided some new clues to the mineralization of mussels.

**Abstract:**

In this study, we cloned a novel matrix protein, cysrichin, with 16.03% homology and a similar protein structure to the coral biomineralized protein galaxin. Tissue expression analysis showed that cysrichin was mainly expressed in mantle and gill tissues. In situ hybridization indicated that cysrichin mRNA was detected in the entire epithelium region of mantle tissue. RNAi analysis and shell notching experiment confirmed that cysrichin participates in the prismatic layer and nacreous layer formation of the shell. An in vitro crystallization experiment showed that the cysrichin protein induced lotus-shaped and round-shaped crystals, which were identified as vaterite crystals. These results may provide new clues for understanding the formation of vaterite in freshwater shellfish.

## 1. Introduction

Mollusks are known for their ability to produce elegant pearls. *Hyriopsis cumingii* is a unique freshwater mussel used for commercial freshwater pearl production in China. The freshwater pearl industry of China accounts for more than 90% of the world’s freshwater pearl production. In recent years, *H. cumingii* has become a good experimental model for studying the biomineralization of freshwater shellfish, and several shell matrix protein genes associated with biomineralization have been reported [1]. Calcium carbonate is the most common material used by mollusks to construct their shells; the mollusk shell can effectively protect the viscera and organism of the mollusk [2]. The shell comprises 95% (*w*/*w*) calcium carbonate and only 5% (*w*/*w*) organic matrix, which includes polysaccharides, matrix proteins, and lipids [3]. The organic matrix of mollusks contains shell matrix proteins, polysaccharides, and a small number of lipids, among which shell matrix proteins play a crucial role in the formation of shells and pearls [4,5]. The mollusk shell has recently become the focus of research in biomaterial science because of its excellent biomechanical properties, such as high toughness and strength [6,7]. Calcium carbonate present in the shell is also being increasingly applied in the pharmaceutical industry.

The shell is composed of two calcified layers, namely, a prismatic layer and a nacreous layer, and its formation is controlled by shell matrix proteins. Soluble matrix proteins can control the nucleation and growth of calcium carbonate crystals; the shell matrix proteins are related to the specific crystal phase and control the transition between aragonite and calcite [8,9,10]. The acidic matrix protein Pif 80 can enrich calcium carbonate, induce aragonite crystal formation, and regulate the c-axis direction [11]. Several matrix proteins of *H. cumingii are* rich in one or more amino acid residues, enrichment regions and repeats of specific amino acids, a chitin-binding domain, and a Ca^2+^ binding site [12]. In other words, matrix proteins play a critical role in shell biomineralization. The calcareous skeleton of coral is very similar to that of humans; hence, it has been used as a human bone graft substitute [13]. Fukual et al. [14] purified the galaxin protein first from the calcified exoskeleton of the reef coral *Galaxea fascicularis* and revealed that it had a characteristic tandem repeat structure. The galaxin-related gene Amgalaxin is involved in the organic skeleton formation of the scleractinian coral *Acropora millepora* [15]. Subsequent studies have revealed that galaxin-like proteins have high serine content and a tandem repeat structure and may have calcium-binding activity, which may be similar to those of mucins; thus, these proteins may have evolved from mucin-type ancestors [16]. Therefore, the galaxin protein is an important class of mineralized proteins.

In the present study, cysrichin, a potential matrix protein, was cloned from the cDNA library of *H. cumingii*. Tissue expression analysis showed that cysrichin was strongly expressed in mantle tissue; furthermore, cysrichin mRNA was detected in the entire epithelium region of mantle tissue by in situ hybridization (ISH). These results provided new insights into how matrix proteins regulate calcium carbonate crystal growth in freshwater shellfish.

## 2. Materials and Methods

### 2.1. Biological Material

Total RNA was extracted with RNAiso Plus (Takara, Japan), and the concentration was determined using a NanoDrop ND 2000 spectrophotometer (Thermo Scientific, Waltham, MA, USA). Nacrein was first found in *Pinctada fucata*, and it contains the CA domain [17]. As its complete sequence from *H. cumingii* is yet to be cloned, we decided to clone its cDNA sequence from the mantle. On the basis of the amino acid sequence “DAGFS,” we designed two 3′-RACE primers, namely, cysrichin-F1 (5′-GAYGCNGGNTTYAGY-3′, Y = C/T, N = A/G/C/T) and cysrichin-F2 (5′-GAYGCNGGNTTYTCN-3′, Y = C/T, N = A/G/C/T). The 3′ RACE polymerase chain reaction (PCR) was performed using the Advantage 2 cDNA polymerase mix according to the manufacturer’s instructions. Based on the 3′ RACE product, the specific primer cysrichin-R1 was designed for 5′ RACE. Following PCR, the 5′ RACE product was sequenced by Sangon Biotech (Shanghai, China) Co., Ltd.

### 2.2. Sequencing Analysis of Cysrichin

BLAST (https://blast.ncbi.nlm.nih.gov/Blast.cgi, accessed on 20 October 2022) was used to analyze the cDNA nucleotide and amino acid sequence homology of cysrichin. The open reading frame (ORF) of cysrichin was deduced by ORF Finder (www.ncbi.nlm.nih.gov/orffinder, accessed on 20 October 2022). The signal peptide of cysrichin was predicted by SignalP (http://www.cbs.dtu.dk/services/SignalP/, accessed on 20 October 2022). The protein parameters were computed using the ExPasy website (http://web.expasy.org/protparam/, accessed on 20 October 2022). The protein’s secondary structure was analyzed by Phyre2 (www.sbg.bio.ic.ac.uk/phyre2/html/page.cgi?id=index, accessed on 20 October 2022).

### 2.3. Tissue Expression Analysis of Cysrichin

RNA reverse-transcription was performed using a PrimeScript™ RT reagent kit with gDNA Eraser (Takara, Japan) at the final cDNA concentration of 5 ng/µL. Table 1 shows the specific primers (qPCR-F and qPCR-R) of cysrichin and primers (EF-1α-F and EF-1α-R) of the internal reference gene *EF-1*. qRT-PCR was performed in a reaction mixture containing 10 µL 2×SYBR qPCR super mix, 0.8 µL of each primer, 1.6 µL cDNA, and 6.8 µL RNase-free water. The PCR conditions were as follows: 94 °C for 5 min, followed by 95 °C for 15 s, 58 °C for 45 s (35 cycles), 95 °C for 15 s, and 65–95 °C for the dissociation curve analysis. Finally, the relative expression levels of cysrichin in various tissues were calculated using the 2^−δδct^ method. The results of RT-PCR and qRT-PCR (repeated six times per sample) were expressed as mean ± SE and were analyzed by one-way ANOVA in SPSS version 18.0 (*p* < 0.05 was considered significant) (Appendix A).

### 2.4. Determination of Cysrichin Location by Using a DNA Probe

A 222 bp fragment (nucleic acid sequence at position 45–266 in Figure 1) with adequate specificity was selected and synthesized by Sangon Biotech (Shanghai, China) Co., Ltd. After nucleotide synthesis, the 222 bp DNA probe was labeled using the DIG DNA labeling mix (Roche, Switzerland). Mantle tissue was removed and immediately fixed with 4% paraformaldehyde for 5 h, followed by dehydration with 20% sucrose solution at 4 °C for 12 h. Subsequently, 10 µm thick tissue sections were obtained using a freezing microtome. Tissue hybridization was performed using a DIG nucleic acid detection kit (Roche, Switzerland). Finally, mantle tissue was photographed when the hybridization signal was observed.

### 2.5. RNAi Assay

We designed three double-stranded RNA (dsRNA) sequences based on the ORF and selected the one with the best inhibitory effect. While designing dsRNA sequences, two templates were first synthesized by two pairs of specificity primers (RNAi-F+T7 and RNAi-R, RNAi-R+T7, and RNAi-F), as shown in Table 1. The two templates were then transcribed with T7 RNA polymerase (Takara, Japan). The dsRNA of cysrichin was synthesized by mixing the two transcription products. The obtained dsRNA was diluted to 0.4 µg/µL with phosphate-buffered saline (PBS), and 100 µL of the diluted dsRNA was injected into the adductor muscle of 1-year-old *H. cumingii* (10 individuals). The control group was injected with PBS. Mantle tissues were collected, and qRT-PCR was performed after 7 days. The shell was gently cleaned, and a 1 × 1 cm portion was cut from the edge area to observe the crystal form by using a scanning electron microscope (Zeiss Sigma 300, Zeiss, Oberkochen, Germany).

### 2.6. Cysrichin Function in Shell Regeneration

Mollusks can automatically repair their shell after artificial damage. To determine the function of cysrichin in shell regeneration, a V-shape notch was cut near the shell margin without injuring mantle tissue. After the operation, mantle tissue was immediately collected as the control group, and the mussels were allowed to continue their growth. Mantle tissue adjacent to the notch was collected at 0, 12, 24, and 48 h and at 4, 7, 11, 15, 20, 25, and 30 d. The mantle tissue of normal *H. cumingii* was simultaneously collected as the control group. RNA extraction and reverse-transcription were performed as described in Section 2.3, and the cysrichin gene expression level was detected by qRT-PCR.

### 2.7. In Vitro Crystallization Experiment

The cysrichin polypeptide was synthesized by ChinaPeptides Co., Ltd. (Shanghai, China), and the amino acid sequence was “QWGDAVCGHSRYSPAFSMCCNGVVQSKSGLEPACCGTR”. The cysrichin polypeptide was diluted to 60 and 90 µg/mL and then mixed with a saturated calcium bicarbonate solution to study its function in crystal morphology. The crystallization reaction system was 20 µL, and pure saturated calcium carbonate solution was used as the control group. Crystallization was performed on cover glasses (Fisherbrand, Pittsburgh, PA, USA), which were dried naturally at room temperature for 48 h. The products were subsequently sent to the Shanghai Moyan Testing Technology Center to detect crystal morphology. Raman spectroscopy for characterizing the crystal was performed at the instrumentation and service center for molecular sciences of Westlake University (Hangzhou, China).

## 3. Results

### 3.1. In Silico Analysis of Cysrichin

According to the results of Blast analysis, the nucleic acid and protein sequences of cysrichin showed no significant homology to known proteins; thus, cysrichin is a novel protein. The full-length cDNA sequence of the cysrichin gene was 648 bp, with a 492 bp ORF encoding 163 amino acids, and the sequence of the first 26 amino acids is thought to represent a hypothetical signal peptide (Figure 1). The cysrichin protein is rich in Cys (14.7%), Ser (11.7%), and Gly (8.64%). According to ExPasy prediction, the theoretical molecular weight of cysrichin was 17.53 kDa, and the theoretical isoelectric point was 8.15 (Table 2). The sequence of the cysrichin protein showed 16.03% consensus with that of the galaxin protein; however, the amino acid sequence of cysrichin accounts for only a part of galaxin, and sequence alignment did not contain amino acids at positions 1–50 and 221–343 of galaxin (Figure 2). The cysrichin protein has five tandem repeat structures (-CG—Y—FSMCC—S-SGL-PA—represents an arbitrary amino acid) and has di-cys motifs per repeat unit; a similar tandem repeat structure and di-cys motifs are found in the galaxin protein (Figure 3). ProtScale assay revealed the cysrichin protein has a hydrophobic region in the N-terminal and several strongly hydrophilic regions in the C-terminal. The hydrophobic blocks and hydrophilic blocks of the entire protein are arranged alternately. The secondary structure prediction indicated that disordered and β-sheet structures accounted for 24% and 42%, respectively (Figure 4). Based on the hydrophobicity analysis and secondary structure prediction, we suspected that cysrichin might induce vaterite formation.

### 3.2. Cysrichin Expression Analysis and Location

The relative expression level of the cysrichin gene in different tissues is shown in Figure 5A. Cysrichin was strongly expressed in the mantle and showed a low expression level in the gill, adductor muscle, and foot. ISH results showed that hybridization signals were detected in epithelial cells in both the dorsal edge and center region of the mantle (Figure 5B).

### 3.3. Gene Silencing and Shell Surface Observation

The dsRNA can specifically bind to the target mRNA and fragment it, thereby blocking the expression of the target protein. In this study, we injected the cysrichin-dsRNA into *H. cumingii*, and the cysrichin expression level in the mantle was investigated after 7 days. The results showed that the cysrichin expression level was reduced by 70% after cysrichin-dsRNA administration (Figure 6i). The surface growth was affected in the *H. cumingii* shell treated with dsRNA. In the nacreous layer, the size of aragonite crystals was not significantly changed; however, the growth of the crystal edges significantly deteriorated (Figure 6B). Compared to the control group, the prismatic surface was not smooth, and the prism crystal growth was irregular (Figure 6D). Moreover, the inhibition of cysrichin expression led to irregular granular deposits on the surfaces of both calcified layers.

### 3.4. Cysrichin Expression during Shell Regeneration

Figure 7 shows the cysrichin expression pattern after shell notching. Both the experimental and control groups showed a similar expression trend. Thus, the change in the expression trend might be related to the physiological status of mussels. The cysrichin gene expression level was significantly increased to its maximum value on day 2, and a high expression level was maintained from day 2 to day 30. The experimental group showed a higher cysrichin expression level than the control group, except on day 20.

### 3.5. Effect of Cysrichin on Crystallization

The purity of the cysrichin polypeptide was 95%. The results of mass spectrometry showed that the cysrichin polypeptide sequence was correct (Figure 8G), and the purity was consistent with the results of the in vitro crystallization experiment. In the present study, 90 and 60 µg/mL concentrations of the cysrichin peptide were added to the system. The scanning electron microscopy (SEM) results showed that cysrichin induced the deposition of similar lotus-shaped and round-shaped crystals at both concentrations (Figure 8C–F). As shown in Figure 9, the crystal’s characteristic peaks were observed at 112, 267, 303, 748, and 1089 cm^−1^, which were identified as vaterite. In the control group, the crystals showed characteristic peaks at 157, 283, 713, and 1087 cm^−1^, which were identified as calcite. Moreover, no difference was observed in the shape and proportion of the two crystal types in the reaction system with different concentrations of the cysrichin peptide. However, the number of vaterite crystals was greater at a 90 µg/mL concentration than at a 60 µg/mL concentration.

## 4. Discussion

In the present study, we cloned a novel matrix protein—cysrichin—from the cDNA library of *H. cumingii*. Cysrichin was identified as a potential matrix protein candidate through tissue expression analysis and ISH. The tissue expression analysis showed that cysrichin was strongly expressed in the mantle and had low expression in the gill relative to that in the mantle. In shellfish, the mantle secretes matrix proteins to regulate shell formation, the outer epithelium at the outer fold edge of the mantle regulates prismatic layer formation, and the dorsal region controls nacreous layer formation [18]. Cysrichin mRNA was highly expressed in both edge and dorsal regions. Therefore, cysrichin might be a matrix protein associated with the calcification of the nacreous and prismatic layers. Like most mollusks, bivalves have an open circulatory system consisting of hemocytes that circulate in soft tissues [19,20]. Previous studies have shown that hemocytes may be involved in the process of shell formation and regeneration [21]. The cysrichin gene was also found to be expressed in gill tissue, thus suggesting that the cysrichin protein might also be transported by hemocytes and is involved in the mineralization process.

Gene-silencing technology is widely used to investigate gene function. The crystal growth of the nacreous layer was severely inhibited, and a cavity appeared in the middle of the crystal after hc-upsalin gene interference [22]. The ultimate role of the matrix protein is to regulate the morphology and orientation of calcium carbonate crystals. In the present study, we treated *H. cumingii* with cysrichin-dsRNA and observed the changes in the shell’s prismatic layer and nacreous layer. Seven days after treatment, cysrichin’s relative expression level in the mantle decreased by 70% compared to that in the PBS treatment group. The normal growth of the crystals in the nacreous and prismatic layer was also affected. The hexagonal morphology of the crystals in the nacreous layer was lost, and the crystal growth in the prismatic layer was irregular (Figure 6). These results indicated that cysrichin might dominate the precise regulation of both the prismatic and nacreous layers.

Disulfide bonds are formed by the covalent linkage between two cysteines that are either on the same protein or on different proteins. Disulfide bonds play a role in the stability and conformational dynamics of peptides and proteins [23,24]. It is predicted that cysrichin is a basic matrix protein, and it is rich in Cys (14.7%), Ser (11.7%), and Gly (8.6%) residues (Table 2). The high content of Cys residues in cysrichin might be related to the formation of disulfide bridges. A surprising result was that the cysrichin gene is clearly similar to the galaxin gene, which is an essential matrix protein found in coral. The cysrichin protein showed 16.03% consensus with the galaxin protein, and the most significant characteristic was the presence of repeated motifs of dicysteine residues. Galaxin is the most abundant and important protein in the coral organic matrix and relates to coral skeleton formation [25]. Framework proteins are frequently Cys-rich [26,27], and the double Cys motif can form intramolecular crosslinks through disulfide bonds [27]. Previous studies have shown that the double cysteine motif may form cyclo-cystine loops [28], which may limit the ability of cysteine residues to form intramolecular (or intermolecular) crosslinks [29]. Therefore, it is probable that only the cysteine residues at the terminal of the protein sequence are available to form intermolecular disulfide bonds [14]. The dicysteine motif is present in the proteins of several mollusks, such as the N14 protein of *Pinctada maxima* [30], lustrin A of *Haliotis rufescens* [27], and pearlin of *Pinctada fucata* [31]. Alejandro et al. [15] concluded that proteins containing di-Cys repeats might be related to the formation of organic skeletons. Dicysteine motifs exist in the cysrichin protein, and there are Cys residues at both terminals of the protein. In addition, several sequence regions in the cysrichin protein have high repeatability. The amino acid sequence “PACCGT” (residues 58–63, 86–91, 114–119, and 142–147) and “SMCC” (residues 43–46, 71–74, 99–102, 127–130, and 155–158). Lengthier amino acid sequences “AYDARFSMCCSDNIQSRSGL” (residues 65–84, 93–112, and 121–140) also existed in the cysrichin protein sequence. The repeat amino acid sequences or these modular structures may be related to the protein’s secondary structure. Hydrophobic analysis showed that the alternating arrangement of hydrophobic and hydrophilic blocks in the entire protein was consistent with the tandem repetition pattern of the protein structure. The protein cysrichin has a hydrophobic region in the N-terminal and several strongly hydrophilic regions in the C-terminal. The strongly hydrophilic regions correspond to the β-sheet structure region of the protein’s secondary structure, thus suggesting that the C-terminal hydrophilic region involved in β-sheet conformation might combine with the organic framework. Therefore, we inferred that cysrichin may be involved in establishing the structural framework of the protein and in inducing nucleation and crystal growth.

Shell formation in mollusks is generally considered to be induced by extracellular events mediated by the organic matrix secreted by mantle epithelium [3]. During shell formation, most of the matrix proteins are integrated into the shell structure after mantle tissue secretion approaches the inner side of the shell [32]. Crystal formation is closely related to complex interactions between organic phases and inorganic ions. In the present study, shell notching was performed to confirm that the cysrichin gene is involved in shell repair, as some known matrix protein genes are reported to be upregulated after deliberate damage to the shell [33]. We found that cysrichin gene expression was higher in the experimental group than in the control group throughout the shell regeneration process (Figure 7). It is reasonable to expect that the matrix protein expression increased in preparation to further accelerate shell deposition. Cysrichin expression remained at a high level after reaching the peak on the fourth day. However, no significant increase in cysrichin expression was observed within 24 h after shell notching. A previous study claimed that calcium carbonate deposited first and then matrix proteins regulated its regular growth [34]. Therefore, we suggest that the cysrichin protein might play a role in the precise deposition of calcium carbonate.

Vaterite is widely used in the pharmaceutical industry because of its large specific surface area, high solubility, high biocompatibility, low specific gravity, and other beneficial characteristics [35,36]. Under normal temperature and pressure, aragonite/vaterite is slightly inferior to calcite in terms of thermodynamic stability [9]. Therefore, calcium carbonate minerals in nature are generally calcite crystals, while calcium carbonate exists in the nacre as aragonite because of the matrix protein. Previous studies have suggested that vaterite may be a transitional stage in nacre formation and can only be discovered when its further transformation is inhibited [37]. Vaterite is an internal factor that affects the quality of freshwater pearls, and a high content of vaterite will deteriorate pearl quality [38]. Hasse et al. [39] found that the content of vaterite was not evenly distributed in the shell and increased gradually from the oldest mineralized part of the shell to the newly mineralized zone. However, vaterite is not found in every matte freshwater pearl; hence, its effect on pearl quality needs further studies. Macías-Sánchez et al. [40] observed a higher content of organic molecules in the amorphous phase than in the crystalline phase and suggested that the final nanogranular structure of nacre is produced by these organic macromolecules. A 48 kDa glycoprotein from the pearl soluble extract might interact with calcium and carbonate ions and induce the formation of vaterite [41]. By performing an ACC transition experiment, Yan et al. found that vaterite could be detected only when its further transformation was inhibited [37]. The specific binding of N25 to the crystal surface may inhibit subsequent transformation to other stable types of polymorphs and lead to the presence of vaterite [42]. In the present study, the cysrichin peptide induced the lotus-shaped and round-shaped crystals, and the Raman spectrum showed characteristic peaks of the crystals at 112, 267, 303, 748, and 1089 cm^−1^. On the basis of previous studies, these crystals were identified as vaterite [43] (pp. 19–20). Cysrichin mainly affects crystal morphology, and the number of vaterite crystals increased with the increase in concentration (from 60 to 90 µg/mL). Thus, we believe that cysrichin could participate in the crystal phase transition during shell formation.

## 5. Conclusions

Here, we reported that the novel matrix protein cysrichin functioned as an establisher of a structural framework of the shell through its special protein structure. Cysrichin has similar sequences and structures to mineralized proteins in other species, such as corals, which can be interpreted from the perspective of biological evolution and links the study of mineralization in mussels to that in corals. No protein that can induce the formation of vaterite has ever been found in *Hyriopsis cumingii*. In this paper, we confirmed the discovery of a protein that can induce the formation of vaterite. The in vitro crystallization experiment implied that cysrichin could inhibit the further transformation of vaterite and induce its final formation, and new members of the vaterite family proteins were amplified. This study extended the research of vaterite in *H. cumingii* and provided some new clues to the biomineralization of mussels.

## Figures and Tables

**Figure 1 biology-12-00447-f001:**
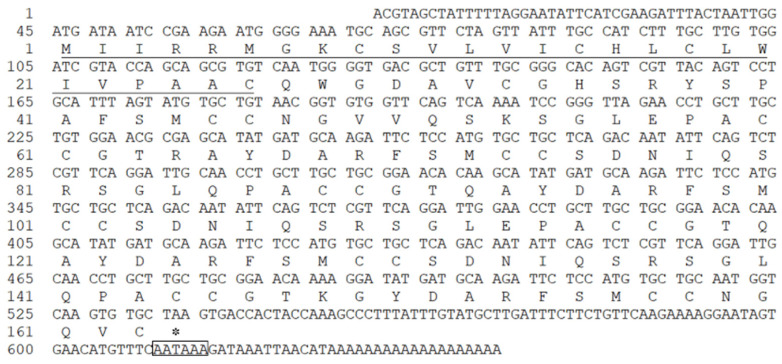
The cDNA and deduced amino acid sequence of cysrichin. Note: the underlined sequence represents a putative signal peptide; the boxed sequence is a putative polyadenylation signal (AATAAA); the cDNA sequence has been deposited in GenBank (Accession No.: MN829946).

**Figure 2 biology-12-00447-f002:**
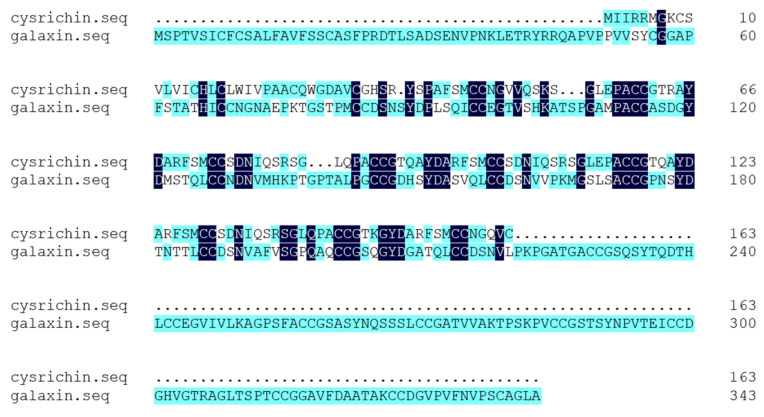
Alignment of amino acid sequences of cysrichin and galaxin. Note: the number on the right represents the position of the last amino acid.

**Figure 3 biology-12-00447-f003:**
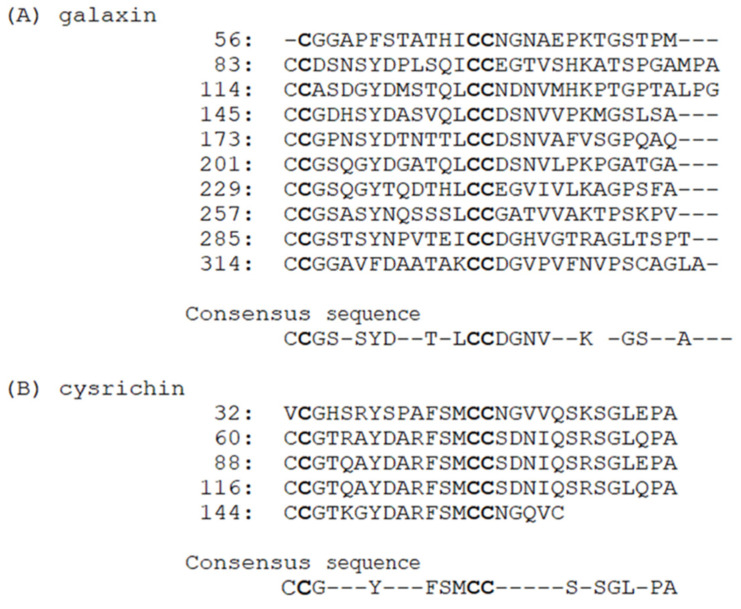
The tandem repeat structure in galaxin and cysrichin. Note: The information about galaxin was obtained from the study of Fukuda et al. [14], GenBank Accession No. AB086183. The galaxin protein was isolated from coral *Galaxea fascicularis*. Numbers on the left indicate the position in the predicted protein, and “−” represents an arbitrary amino acid. The consensus sequence for the repeats from each protein is shown beneath the alignments.

**Figure 4 biology-12-00447-f004:**
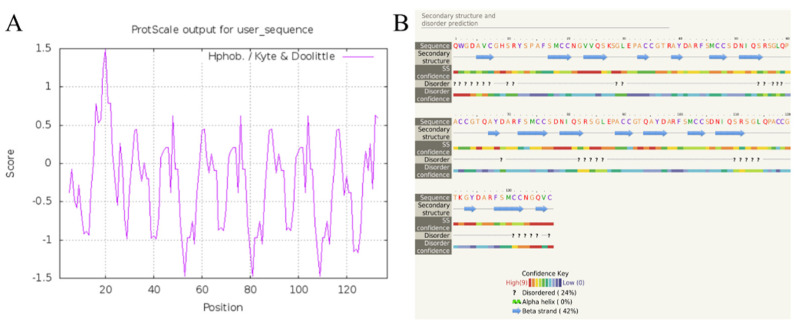
Hydrophobicity analysis (**A**) and secondary structure prediction (**B**) of cysrichin. Note: hydrophobicity analysis was performed by ProtScale, and secondary structure was predicted with Phyre2.

**Figure 5 biology-12-00447-f005:**
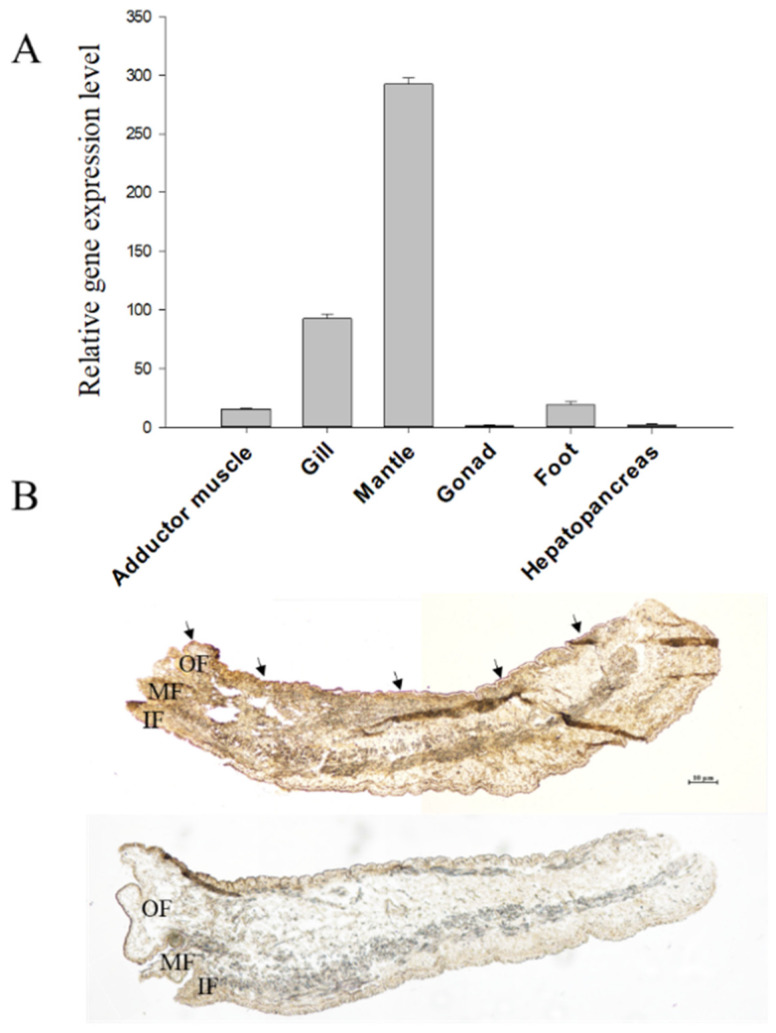
Tissue expression analysis (**A**) and ISH (**B**) results. Note: (**B**), purple hybridization signals were detected in the outer fold epithelial cells of the mantle in the experimental group. OF: outer fold, MF: middle fold, IF: inner fold.

**Figure 6 biology-12-00447-f006:**
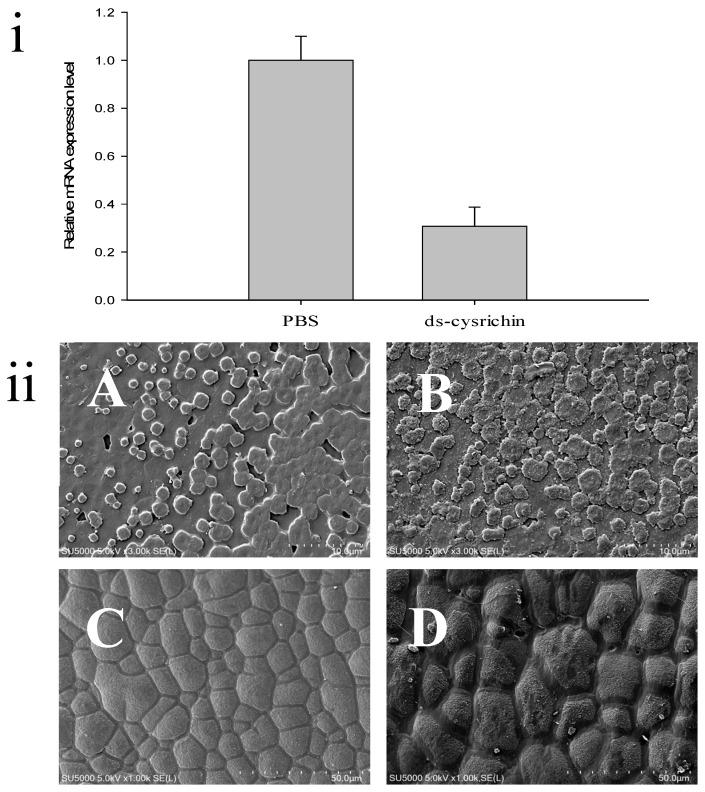
Influence of cysrichin on the morphology of newly formed crystals on the shell. (**i**) The relative expression level of cysrichin after RNAi; (**ii**) Crystal morphology of the shell surface after RNAi. Note: (**A**,**B**) The nacreous layer; (**C**,**D**) The prismatic layer; (**A**,**C**) The PBS group; (**B**,**D**) The ds-cysrichin group.

**Figure 7 biology-12-00447-f007:**
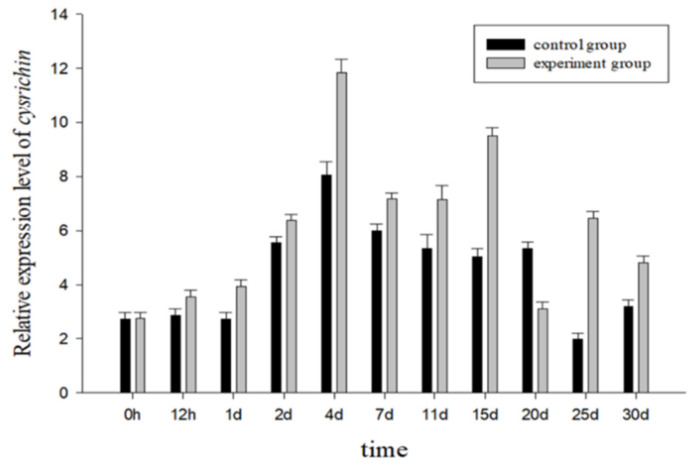
The cysrichin expression pattern from 0 to 30 d after shell notching.

**Figure 8 biology-12-00447-f008:**
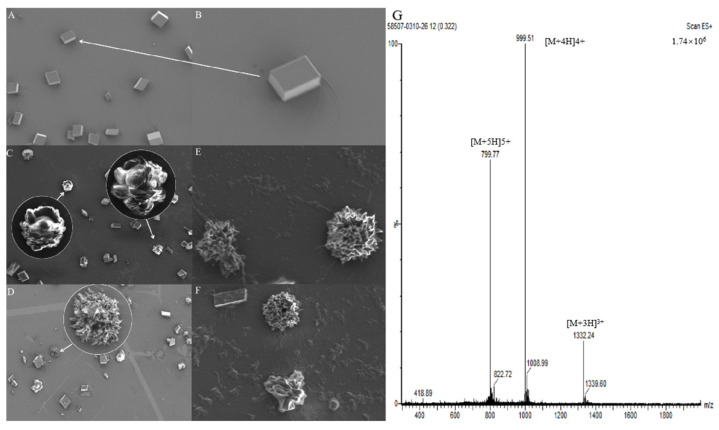
Crystal morphology detection by SEM and MS analysis of the cysrichin polypeptide. Note: (**A**) Control group, (**B**) Magnified view of the crystal indicated by the arrow; (**C**–**F**) Effects of cysrichin on crystallization; the circled areas are the magnified view of the arrow. (**C**,**E**,**F**) When a total of 90 µg/mL cysrichin was added to the system; (**D**) when 60 µg/mL cysrichin was added to the system. (**G**) Mass spectrogram of the synthetic cysrichin peptide.

**Figure 9 biology-12-00447-f009:**
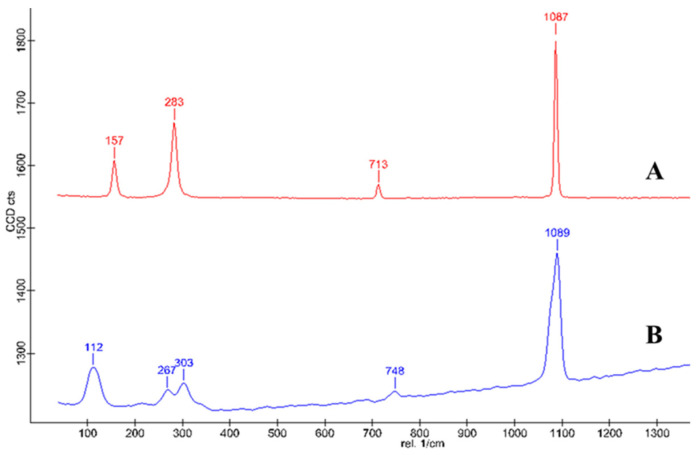
Raman spectrum of the formed crystal. Note—A: crystals in the control group; B: Crystals in the experimental group.

**Table 1 biology-12-00447-t001:** Primers used in this study.

Primer Name	Primer Sequence (5′–3′)
cysrichin-F1	5′-GAYGCNGGNTTYAGY-3′, Y = C/T, N = A/G/C/T
cysrichin-F2	5′-GAYGCNGGNTTYTCN-3′, Y = C/T, N = A/G/C/T
cysrichin-R	GCATACAAATAAAGGGCTTTGGTAG
qPCR-F	GATGATAATCCGAAGAATGG
qPCR-R	GATTTTGACTGAACCACACC
EF-1α-F	GGAACTTCCCAGGCAGACTGTGC
EF-1α-R	TCAAAACGGGCCGCAGAGAAT
RNAi-F	TATGTGCTGTAACGGTGTGGT
RNAi-R	AAAGGGCTTTGGTAGTGGTC
RNAi-F+T7	GGATCCTAATACGACTCACTATAGGTATGTGCTGTAACGGTGTGGT
RNAi-R+T7	GGATCCTAATACGACTCACTATAGGAAAGGGCTTTGGTAGTGGTC

**Table 2 biology-12-00447-t002:** Composition of the cysrichin protein.

Amino Acid	Proportion
Cys (C)	14.70%
Ser (S)	11.70%
Gly (G)	8.60%
Arg (R)	6.70%
Gln (Q)	6.10%
Asp (D)	4.90%
Ile (I)	4.30%
Leu (L)	4.30%
Met (M)	4.30%
Val (V)	4.30%
Pro (P)	3.70%
Asn (N)	3.10%
Phe (F)	3.10%
Tyr (Y)	3.10%
Thr (T)	2.50%
Lys (K)	1.80%
Glu (E)	1.20%
His (H)	1.20%
Trp (W)	1.20%
Theoretical weight: 17.53 kDa	pI: 8.15

## Data Availability

Not applicable.

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
