# Peer review of "The Matrix Protein Cysrichin, a Galaxin-like Protein from Hyriopsis cumingii, Induces Vaterite Formation In Vitro"

_biology, 2023, doi:10.3390/biology12030447_

Round 1

Reviewer 1 Report

Molluscan shell matrix proteins play a significant role in biomineralization. From the freshwater mussel Hyriopsis cumingii, Xia et al, identified a galaxin-like protein coding gene which was named cysrichin. qPCR analysis of cysrichin mRNA expression indicated high levels in mantle and gill tissue, and during shell regeneration as well. Shell growth was affected by dsRNA gene silencing of cysrichin, characterized by irregular prism crystals and granular deposits.

This manuscript follows standard methods for such studies. It is requested that the authors provide some additional points that can help improve the manuscript as well as address the missing critical elements in methods/statistical analysis/results.

Major Comments

Comment 1 : Section 2.1 : Details of cysrichin RACE-PCR lacking: A more detailed description of RACE for amplification of cysrichin is requested from the authors. Degenerate PCR primers cysrichin-F1 / F2: how were they designed? Were the primers based on sequences of other species or previously published literature? 

Please provide details of the RACE kit used. RACE-PCR often uses a combination of gene-specific and RACE kit-specific primers in single/ nested PCR. Was nested PCR used for 3’ RACE, considering the degeneracy of the 3’ RACE primers? The authors are requested to explain the detailed steps of RACE PCR with primers used in each step, and annealing temperatures for each step so that readers can benefit from the details when trying to clone similar gene orthologs from other mollusks/related invertebrates.

Also, It would be great if the authors can mark the binding sites of the cysrichin-F1 / F2/R1 primers in Figure 1

Comment 2: Section 2.5. RNAi assay – Details of the RNAi assay are very sketchy: Authors should add a note on the rationale of selection and design RNAi sequences for cysrichin. Details of the transfection reagent, size/weight of the animals used; number of animals used for RNAi experiments.

Comment 3 : Missing a critical siRNA control to rule out off target RNAi effects: A Non-targeting siRNA/ control or a scrambled RNAi control is an essential control to rule out off-target / non-specific RNAi effects in gene silencing experiments. PBS control cannot be a substitute for this (Figure 6 ) as PBS injection cannot be compared to dsRNA in-vivo injection, considering the cellular response to dsRNA injection. Kindly provide the data for the non-targeting siRNA /scrambled control also for Figure 6.

Comment 4 : Size of the predicted cysrichin protein – Galaxins and galaxin-like-proteins like agalaxins are comparatively larger in size around 250-350 amino acids, whereas cysrichin is only half the size (160 aa). Can the authors provide information that they have cloned the complete cDNA for this gene and that this cloned fragment coding for the shorter protein is not a splice variant or a partial gene sequence? Were the authors able to align the nucleotide sequence to the genomic sequence of Hyriopsis cumingii.

Comment 5: Improve protein alignment -Figure 2- It would be great if the authors were able to include a few more sequences (four or five) of galaxin/amgalaxin or shell matrix protein from molluscs in the amino acid alignment in Figure 2 and indicate the conservation of residues involved. An amino acid alignment with residues color-coded according to their physiochemical properties may be able to better indicate conservation. Additionally, it would be helpful if the authors could improve the figure legends and explain what residues highlighted in blue/black indicate.

Comment 6 : Please include a Phylogenetic tree for better visualizing the evolutionary relationships : To illustrate the evolutionary relationship between Hyriopsis cumingii cysrichin and other matrix proteins such as galaxins, galaxin-like proteins, amgalaxin, other molluscan shell matrix proteins, a phylogenetic tree (based on amino acid sequences) of these proteins would be beneficial.

Comment 7 : Statistical Analysis and significance data 

for relative gene expression: For the relative expression analysis by qPCR ( Figure 6.1 and figure 7), the authors are requested to provide the appropriate statistical details regarding the number of replicates, statistical methods used for analysis, and significance values in the figure legends/results. A paragraph on the statistical analysis in the methods section would also be welcome.

Other  Minor Comments

It is recommended that the authors improve the language and grammar in the manuscript. A few examples

Line 301 -“AYDARFSMCCSD- NIQSRSGL” (residues 65-84, 93-112, and 121-140) also exited in the cysrichin protein - exited should be replaced by existed

Author Response

Reviewer1

Molluscan shell matrix proteins play a significant role in biomineralization. From the freshwater mussel Hyriopsis cumingii, Xia et al, identified a galaxin-like protein coding gene which was named cysrichin. qPCR analysis of cysrichin mRNA expression indicated high levels in mantle and gill tissue, and during shell regeneration as well. Shell growth was affected by dsRNA gene silencing of cysrichin, characterized by irregular prism crystals and granular deposits.

This manuscript follows standard methods for such studies. It is requested that the authors provide some additional points that can help improve the manuscript as well as address the missing critical elements in methods/statistical analysis/results.

Major Comments

Comment 1 : Section 2.1 : Details of cysrichin RACE-PCR lacking: A more detailed description of RACE for amplification of cysrichin is requested from the authors. Degenerate PCR primers cysrichin-F1 / F2: how were they designed? Were the primers based on sequences of other species or previously published literature? 

Please provide details of the RACE kit used. RACE-PCR often uses a combination of gene-specific and RACE kit-specific primers in single/ nested PCR. Was nested PCR used for 3’ RACE, considering the degeneracy of the 3’ RACE primers? The authors are requested to explain the detailed steps of RACE PCR with primers used in each step, and annealing temperatures for each step so that readers can benefit from the details when trying to clone similar gene orthologs from other mollusks/related invertebrates.

Also, It would be great if the authors can mark the binding sites of the cysrichin-F1 / F2/R1 primers in Figure 1

Answer: Thanks for the reviewer’s useful comments.

  • The primers cysrichin-F1 (5’-GAYGCNGGNTTYAGY-3’, Y=C/T, N=A/G/C/T) and cysrichin-F2 (5’-GAYGCNGGNTTYTCN-3’, Y=C/T, N=A/G/C/T) designed to be based on the amino acid sequence “DAGFS”. In fact, there are dozens of forward primers. After 3’ RACE, we got a RNA chain and according to it to design reverse primer. Finally, we got cysrichin full-length cDNA.The detailed steps of RACE PCR were according to the manufacturer's instructions for 5 '/3' RACE Kit (Roche, Switzerland).

Comment 2: Section 2.5. RNAi assay – Details of the RNAi assay are very sketchy: Authors should add a note on the rationale of selection and design RNAi sequences for cysrichin. Details of the transfection reagent, size/weight of the animals used; number of animals used for RNAi experiments.

Answer: Thanks for the reviewer’s useful comments.

  • We designed three ds-RNA sequences in ORF, each one has a different inhibitory effect, but we chose the one with the most inhibitory effect.In addition, ten Hyriopsis cumingii at 1.5 years of age were used for RNAi assay. Each Hyriopsis cumingii was injected with 100 µL ds-RNA and the concentration of ds-RNA was 0.4µg/µL. And the scanning electron microscope model is SIGMA 300 of ZEISS.

Comment 3 : Missing a critical siRNA control to rule out off target RNAi effects: A Non-targeting siRNA/ control or a scrambled RNAi control is an essential control to rule out off-target / non-specific RNAi effects in gene silencing experiments. PBS control cannot be a substitute for this (Figure 6 ) as PBS injection cannot be compared to dsRNA in-vivo injection, considering the cellular response to dsRNA injection. Kindly provide the data for the non-targeting siRNA /scrambled control also for Figure 6.

Answer: 

  • Actually, we injected PBS and dsRNA of green fluorescent protein (GFP) compared to dsRNA in-vivo injection. The effects of PBS and dsRNA-GFP injections were almost identical.

Comment 4 : Size of the predicted cysrichin protein – Galaxins and galaxin-like-proteins like agalaxins are comparatively larger in size around 250-350 amino acids, whereas cysrichin is only half the size (160 aa). Can the authors provide information that they have cloned the complete cDNA for this gene and that this cloned fragment coding for the shorter protein is not a splice variant or a partial gene sequence? Were the authors able to align the nucleotide sequence to the genomic sequence of Hyriopsis cumingii.

Answer: 

  • The information about galaxinin the study of Fukuda et al.[14]. In addition, we have enough evidence that cysrichin is not a, not a splice variant or a partial gene sequence. The cysrichin has an “AATAA” sequence, which was a poly(A) modification at the 3 '-terminal after eukaryotic transcription.

Fig. 1. Nucleotide sequence of a cDNA coding for galaxin (GenBank/EMBL/DDBJ Accession No.: AB086183).

[1] Fukuda, I.; Ooki, S.; Fujita, T.; Murayama, E.; Nagasawa, H.; Isa, Y.; Watanabe, T. Molecular cloning of a cDNA encoding a soluble protein in the coral exoskeleton. Biochemical & Biophysical Research Communications 2003, 304, 11-17.

Comment 5: Improve protein alignment -Figure 2- It would be great if the authors were able to include a few more sequences (four or five) of galaxin/amgalaxin or shell matrix protein from molluscs in the amino acid alignment in Figure 2 and indicate the conservation of residues involved. An amino acid alignment with residues color-coded according to their physiochemical properties may be able to better indicate conservation. Additionally, it would be helpful if the authors could improve the figure legends and explain what residues highlighted in blue/black indicate.

Comment 6 : Please include a Phylogenetic tree for better visualizing the evolutionary relationships : To illustrate the evolutionary relationship between Hyriopsis cumingii cysrichin and other matrix proteins such as galaxins, galaxin-like proteins, amgalaxin, other molluscan shell matrix proteins, a phylogenetic tree (based on amino acid sequences) of these proteins would be beneficial.

Answer: 

5,6) Cysrichin has the highest homology with galaxin, and galaxin is the earliest protein cloned from coral. Both of them play important roles as organic matrix. The comparison between cysrichin and galaxin can be taken as a small point of view in terms of evolution. But a Phylogenetic tree may not be necessary because there aren't a lot of similar proteins in this special conservative structure, and there isn't a lot of consistent data to support this work.

Comment 7 : Statistical Analysis and significance data for relative gene expression: For the relative expression analysis by qPCR ( Figure 6.1 and figure 7), the authors are requested to provide the appropriate statistical details regarding the number of replicates, statistical methods used for analysis, and significance values in the figure legends/results. A paragraph on the statistical analysis in the methods section would also be welcome.

Answer:

  • The RT-PCR and qRT-PCRrepeated six times per sample. The data form RT-PCR and qRT-PCR were expressed as the means ± SE, and were analyzed in SPSS version 18.0 by one-way ANOVA (p< 0.05 was considered significant). We have included these in Section 2.3 of the manuscript.

Other Minor Comments

It is recommended that the authors improve the language and grammar in the manuscript. A few examples

Line 301 -“AYDARFSMCCSD- NIQSRSGL” (residues 65-84, 93-112, and 121-140) also exited in the cysrichin protein - exited should be replaced by existed

Answer:

 Our manuscript has been subjected to language revision by a professional organization.

Line 301: “-AYDARFSMCCSD- NIQSRSGL (residues 65-84, 93-112, and 121-140) also exited in the cysrichin protein” has been replaced by “-AYDARFSMCCSD- NIQSRSGL (residues 65-84, 93-112, and 121-140) also existed in the cysrichin protein”.

Reviewer 2 Report

Here, Xia and co-authors describe a new shell matrix protein from the freshwater mussel Hyriopsis cumingii that induces vaterite formation in vitro.

COMMENTS

Title:

-       The title is misleading. It should be replaced by “The matrix protein cysrichin, a galaxin-like protein from Hyriopsis cumingii, induces vaterite formation in vitro” in order to accurately reflect the findings of the paper. In addition, species name should be in italic.

Summary and Abstract:

-       These two sections need re-writing since some text passages are not clear.

 Introduction:

-       Given previous thorough works on H. cumingii shell proteins and transcripts (see Berland et al 2013 or Xen et al 2021), it is not clear from the introduction if this is a completely novel protein sequence or if it has been found previously. A comparison with previous works should be given here or in the discussion.  

-       Lines 49-50: It is very simplistic to affirm that the mollusk shell comprises 95% of calcium carbonate and 5% of organic matrix. First, it is not clear from the text if these percentages are in weight, volume, molar, etc. Second, many bivalve species, if not the majority, have less than 1% of the total weight in organic matrix including the Hyriopsis cumingii close relative Unio pictorum (see Marie et al., 2007 and Ramos-Silva et al., 2012).

-       Lines 62-63: Subject missing in the sentence. Please revise the English.

-       Line 78: ISH – lacks the definition.

 Materials and Methods:

-       This section lacks a description of the number of individuals and replicates used in every experiment.

           -       Line 86: The authors wanted to clone nacrein but ended up cloning another sequence. Was this by chance? This is not very-well explained by the authors and it should be clarified.

-       Details of the SEM are missing, including sample preparation and description of the microscope.

Results:

-       Line 157: The word bioinformation is not meaningful. It would be better to replace it by ‘in silico analysis’.

 -       Lines 194-199: Figure 5 B is insufficiently described, how is the signal interpreted? Is it in darker color in the tissue? And, what are the differences between the upper and the lower image?

-       Line 205 – remove ‘making it useless’

 -       More information is needed to interpret Figure 6. Was this experiment done in more than one individual? Are the granules present in the entire surface of the shell or just in the regions that were imaged for Figure 6? More SEMs photos should be provided as supplementary data to inform about the effects on crystal morphology, including one photo of the entire sample with locations where the more detailed SEMs were taken from. Images of the shell wall interior would also be useful to see what is the effect beyond the shell surface.

-       Line 223 significantly.

-       Information on the size of the groups and supplementary raw data used for Figures 5A, 6A and 7 is needed in order to give meaning to the error bars and support the findings of this manuscript.

 Discussion:

-       Lines 324-326: The speculation of a role of cysrichin in CaCO3 deposition in relation to Figure 7 is not sufficiently clear. Furthermore, results on gene silencing (Fig 6) are not clearly addressed in the discussion or used to hypothesize the function of the protein.

 References:

-       This section needs to be checked – for example species names are not italicized.

The manuscript should be entirely revised to correct the English in terms of grammar, plural/singular expressions and typos.

Author Response

Reviewer2

Here, Xia and co-authors describe a new shell matrix protein from the freshwater mussel Hyriopsis cumingii that induces vaterite formation in vitro.

COMMENTS

Title:

-       The title is misleading. It should be replaced by “The matrix protein cysrichin, a galaxin-like protein from Hyriopsis cumingii, induces vaterite formation in vitro” in order to accurately reflect the findings of the paper. In addition, species name should be in italic.

 Answer: Thanks for the reviewer’s useful comments.

According to your suggestion, we will change the title to “The matrix protein cysrichin, a galaxin-like protein from Hyriopsis cumingii, induces vaterite formation in vitro”.

Summary and Abstract:

-       These two sections need re-writing since some text passages are not clear.

 Answer:

We have made corresponding modifications to this part.

 Introduction:

-       Given previous thorough works on H. cumingii shell proteins and transcripts (see Berland et al 2013 or Xen et al 2021), it is not clear from the introduction if this is a completely novel protein sequence or if it has been found previously. A comparison with previous works should be given here or in the discussion.  

-       Lines 49-50: It is very simplistic to affirm that the mollusk shell comprises 95% of calcium carbonate and 5% of organic matrix. First, it is not clear from the text if these percentages are in weight, volume, molar, etc. Second, many bivalve species, if not the majority, have less than 1% of the total weight in organic matrix including the Hyriopsis cumingii close relative Unio pictorum (see Marie et al., 2007 and Ramos-Silva et al., 2012).

-       Lines 62-63: Subject missing in the sentence. Please revise the English.

-       Line 78: ISH – lacks the definition.

Answer:

Lines 49-50: we will change the sentence to “Several matrix proteins of Hyriopsis cumingii were summarized found that the matrix proteins have characteristics rich in one or more amino acid residues...”.

Lines 62-63: The sentence was changed to “Several matrix proteins of H. cumingii are rich in one or more amino acid residues, enrich-ment regions and repeats of specific amino acids, a chitin-binding domain, and a Ca2+ binding site”.

Line 78: The definition of ISH is in situ hybridization, we added this information in the article

 Materials and Methods:

-       This section lacks a description of the number of individuals and replicates used in every experiment.

-       Line 86: The authors wanted to clone nacrein but ended up cloning another sequence. Was this by chance? This is not very-well explained by the authors and it should be clarified.

-       Details of the SEM are missing, including sample preparation and description of the microscope.

Answer:

A description of the number of individuals and replicates used in each experiment has been added.

Line 86: Because nacrein gene does not exist in Hyriopsis cumingii, we cloned a new gene in Hyriopsis cumingii according to the principle of homology, and finally cloned cysrichin.

“Clean the shell gently and cut out the size of 1*1 cm at the edge area to observe the crystal form. The crystals were enlarged by scanning electron miceoscopy (ZEISS, SIGMA 300).” has been added.

Results:

-       Line 157: The word bioinformation is not meaningful. It would be better to replace it by ‘in silico analysis’.

-       Lines 194-199: Figure 5 B is insufficiently described, how is the signal interpreted? Is it in darker color in the tissue? And, what are the differences between the upper and the lower image?

-       Line 205 – remove ‘making it useless’

-       More information is needed to interpret Figure 6. Was this experiment done in more than one individual? Are the granules present in the entire surface of the shell or just in the regions that were imaged for Figure 6? More SEMs photos should be provided as supplementary data to inform about the effects on crystal morphology, including one photo of the entire sample with locations where the more detailed SEMs were taken from. Images of the shell wall interior would also be useful to see what is the effect beyond the shell surface.

-       Line 223 significantly.

-       Information on the size of the groups and supplementary raw data used for Figures 5A, 6A and 7 is needed in order to give meaning to the error bars and support the findings of this manuscript.

Answer:

Line 157: The word ‘bioinformation’ was replace it by ‘in silico analysis’.

Lines 194-199: After probe hybridization, the specific probe will bind to the mRNA of cysrichin. The probe is labeled by DIG and will be stained to produce purple color. There were purple hybridization signals in the outer fold epithelial cells of the mantle in the experimental group.

Line 205: 'making it useless' has been deleted.

Figure 6 shows the morphology of the newly formed calcium carbonate crystals. Inhibition of cysrichin does not affect the morphology of the shell that has been formed. When cysrichin is inhibited, the morphology of the newly formed crystals will be greatly changed in the absence of cysrichin

 Discussion:

-       Lines 324-326: The speculation of a role of cysrichin in CaCO3 deposition in relation to Figure 7 is not sufficiently clear. Furthermore, results on gene silencing (Fig 6) are not clearly addressed in the discussion or used to hypothesize the function of the protein.

Answer: 

In the whole process of shell regeneration, cysrichin gene expression in the experimental group was higher than that in the control group, indicating that it was related to shell regeneration. Then, based on the time when cysrichin expression reached its peak, we speculated that its function might not be related to early nucleation but to precise deposition of calcium carbonate.

And results on gene silencing (Fig 6) are mentioned in the second paragraph of the discussion.

References:

-       This section needs to be checked – for example species names are not italicized.

The manuscript should be entirely revised to correct the English in terms of grammar, plural/singular expressions and typos.

Answer: 

We revised the species names to use italics. 

Our manuscript has been subjected to language revision by a professional organization

Round 2

Reviewer 1 Report

In the revised manuscript, the authors have provided satisfactory explanation to the queries in the first round of review. Details of RACE, RNAi etc, statistical analysis of the gene expression etc is updated in the revised version of the manuscript. I recommend the updated manuscript to be accepted.

Author Response

Thanks for your work and time!

Reviewer 2 Report

The manuscript by Xia and co-authors was substantially improved, but some of the reviewer's comments were not sufficiently addressed to envisage acceptance:

-       Comment on lines 49-50 was not addressed. 

-       Line 86: The answer of the authors should be added to the manuscript to make clear how they found cysrichin. 

-       Comment  "Information on the size of the groups and supplementary raw data used for Figures 5A, 6A and 7 is needed in order to give meaning to the error bars and support the findings of this manuscript."  was not addressed by the authors. The authors must provide supplementary raw data of their experiments to support graphs 5A, 6A and 7.

Author Response

Thanks for the helpful comments.

-lines 49-50:   Comment on lines 49-50 was not addressed.

Answer:The percentages are in weight and It has also explained in the text.

line 86:The answer of the authors should be added to the manuscript to make clear how they found cysrichin.

Answer: According to the amino acid sequence "DAGFS" of Nacrein in Pinctada fucata, we designed degenerate primers F1 (5’-GAYGCNGGNTTYAGY-3’, Y=C/T, N=A/G/C/T) and F2 (5’-GAYGCNGGNTTYTCN-3’, Y=C/T, N=A/G/C/T) for 3 'RACE. A nucleic acid sequence was found in the PCR product according to the next-generation sequencing method, it was named cysrichin. This is also explained in Section 2.1.

Comment  "Information on the size of the groups and supplementary raw data used for Figures 5A, 6A and 7 is needed in order to give meaning to the error bars and support the findings of this manuscript."  was not addressed by the authors. The authors must provide supplementary raw data of their experiments to support graphs 5A, 6A and 7.

Answer: Due to instrument maintenance, we only found the original data of tissue quantification and provided it as the attachment
